# Comparing Performance Between Different Implementations of ROS for Unity

Jordan Allspaw
University of Massachusetts Lowell
Lowell, Massachusetts, USA
Jordan_Allspaw@uml.edu

Gregory LeMasurier
University of Massachusetts Lowell
Lowell, Massachusetts, USA
gregory_lemasurier@student.uml.edu

Holly Yanco
University of Massachusetts Lowell
Lowell, Massachusetts, USA
holly@cs.uml.edu

## ABSTRACT

The Virtual and Augmented reality market has greatly expanded in the recent years with devices such as the Meta Quest and the Microsoft Hololens. Both of these devices have limited to no support on Linux distributions, which are commonly used in robotics. Instead, developers are typically running these on Windows, often with either Unity or Unreal for 3D support, and using a middleware to integrate with other systems such as ROS.

There have been several different approaches for this communication; it can be difficult for developers of new projects to choose one that might be best for their use case. In this paper, we set out to benchmark the performance of several different Unity to ROS communication implementations. In particular we are looking at an analysis on ROS#, Unity Technology's ROS-TCP-Connector, and ROS.NET.

## CCS CONCEPTS

• **Software and its engineering** → **Software performance**.

## KEYWORDS

Robot Operating System (ROS), Unity, ROS-TCP-Connector, ROS.NET, ROS#

**ACM Reference Format:**
Jordan Allspaw, Gregory LeMasurier, and Holly Yanco. 2023. Comparing Performance Between Different Implementations of ROS for Unity. In *Proceedings of HRI-23 Workshop on Virtual, Augmented, and Mixed Reality for Human-Robot Interaction (Virtual, Augmented, and Mixed Reality for HRI).* ACM, New York, NY, USA, 6 pages. https://doi.org/10.1145/nnnnnnn.nnnnnnn

## 1 INTRODUCTION

The consumer VR/AR market has been a major asset to research, providing not only several iterations of hardware, but also many useful tools and applications to be used specifically within the VR/AR space. One tool used heavily within the VAM-HRI domain is Unity, which allows for rapid development and sharing of many kinds of interfaces. Of these projects, many use ROS [28] to communicate data from robots to the Unity project. There are several different Unity-ROS communication methods that groups use within the

VAM-HRI domain. Each of these solutions has different amounts of support, feature-sets, and capabilities, so choosing one for a project can be difficult. Beyond those differences, the choice could also be impacted by the performance of the actual message passing from Unity to the ROS ecosystem. While some applications might only send small amounts of data, VR/AR applications can be very performance intensive, and often require transmitting large amounts of data for visualization. Furthermore, VR/AR applications can be computationally expensive, and so CPU overhead could also be a concern.

Our lab created a Unity-ROS implementation, ROS.NET [2], in 2016, and we have used it for a number of projects ranging from touch screen to VR applications. The authors have no financial interest or other benefits regarding the results presented in this paper. Since our lab developed this implementation in 2016, several alternative implementations have been released. As the field has advanced, we are motivated to investigate the current state of other Unity-ROS approaches to determine whether it would be worth switching implementations for current or future projects. After looking through the implementation that each Unity-ROS approach uses, we believed that there could be trade-offs for each. We thought this information may also be applicable to the VR/AR community and so decided to run a controlled set of benchmarking tests to compare performance under a few different practical situations. We are particularly interested in the effect message size has on latency, publishing rate, and CPU usage for each approach. In addition, we will also check whether publishing to multiple machines has any effect on performance for each approach. We are interested in publishing to multiple machines as one interface developed in Unity might be used to control swarms or other multi-robot scenarios.

## 2 DIFFERENT IMPLEMENTATIONS

In this paper we are comparing the performance of the different Unity-ROS communication implementations through a series of tests explained below. In particular we are looking at an analysis on ROS# [31], Unity Technology's ROS-TCP-Connector [35], and ROS.NET [2]. Each of these implementations has been utilized in the general robotics community, as well as specifically in the VR/AR space. Our justification for including ROS# and ROS-TCP-Connector was based on a small literature review to analyze the popularity of each ROS communication approach within the VAM-HRI community.

For this literature review, we targeted a modern representation of the software used, thus we only include papers published from 2020 until January 10th 2023. This range was chosen as ROS-TCP-Connector was first publicly released in 2020 [34]. We began the literature review by including all VAM-HRI papers that meet this

| ROS | Count | Used By |
|---|---|---|
| ROS# | 15 | [4–6, 9, 10, 13–17, 23, 25–27, 42] |
| ROS-TCP-Connector | 5 | [8, 19, 39–41] |
| rosbridge | 5 | [3, 7, 12, 32, 33] |
| ROS Reality | 5 | [11, 20, 29, 30, 37] |
| ROS.NET | 1 | [18] |
| ROSIntegration | 1 | [43] |

**Table 1: Popularity of different Unity-ROS implementations within the VAM-HRI community since 2020.**

criteria. Next we included all references of the VAM-HRI papers that also fit the same criteria. Finally, we conducted a forward search rooted from the VAM-HRI papers using the citations listed on Google Scholar. Each branch of the forward search continued until we encountered a paper that did not have any citations or that did not include the key terms "ROS" or "Robot Operating System" within their paper or linked Github repositories. We conducted our forward search in this manner as some of the papers that did not fit the criteria for our forward search had thousands of citations stemming from them, most of which were not within our domain.

Once all of the papers were collected, we first removed all papers that did not include the key terms "ROS" or "Robot Operating System" within their paper or linked Github repositories. This left us with a total of 56 papers that fit our criteria. Out of these papers, 24 either did not use ROS on Windows or did not specify the method they used to do so. Out of the remaining 32 papers, 15 used ROS#, 5 used ROS-TCP-Connector, 5 used rosbridge [21], 5 used ROS Reality [38], 1 used ROS.NET, and 1 used ROSIntegration [22]. We excluded ROS Reality from our analysis as the creators no longer support it as they have migrated to ROS# [1]. Additionally we have excluded ROSIntegration as it is only used with Unreal Engine, not Unity. Table 1 breaks down the papers included in each category.

## 2.1 ROS#

ROS# was developed by Dr. Martin Bischoff and "is a set of open source software libraries and tools in C# for communicating with ROS from .NET applications"[31] It uses a rosbridge server for communicating with ROS, which uses JSON to allow communication. This approach requires a rosbridge server node to be running on a machine in the ROS environment, that receives data from Unity and republishes it as a ROS message. As such, all ROS communication to Unity first passes through this rosbridge node. This could be a potential bottleneck in multi-machine environments. This methodology potentially reduces CPU usage on the computer running the Unity application as the serialization is offloaded to another computer.

This is one of the earlier and most widely used approaches in the VR/AR community, and has served several projects as shown through our literature review. There are several other Unity applications that also use rosbridge for the actual ROS communication such as ROS Reality developed by the Humans to Robots Lab at Brown University. However ROS Reality is only supported for legacy projects according to [1], which recommends switching to ROS#. Since ROS Reality also uses rosbridge, we expect that it would have similar performance to ROS#. Finally, some some papers in

our survey utilized rosbridge by itself without a suite, or did not name the suite, but as the underlying communication layer is the same. We chose to test the most popular implementation, ROS#, to cover all rosbridge approaches.

## 2.2 ROS-TCP-Connector

Unity Robotics Hub was created and is maintained directly by a team at Unity as part of their robotics initiative to allow for communication from within Unity to ROS. They use ROS-TCP-Connector [35] for communication to ROS, which is another approach built on rosbridge. They also credit ROS#, although they maintain their own separate approach. This requires a ROS-TCP-Endpoint [36] running on a machine in the ROS environment. This node receives data from Unity and translates it to ROS. Likewise, all ROS communication to Unity first passes through this connector. While this could be a potential bottleneck in multi-machine environments, it might also reduce CPU usage for the computer running the Unity application as it is being offloaded to another computer. Despite being newer than other implementations discussed here, it has already seen adoption for projects within the VR/AR community, as seen in our literature review. Additionally, this implementation's backing by Unity's robotics initiative means that is it professionally maintained and supported.

## 2.3 ROS.NET

ROS.NET [2] was created by our lab to allow communication between windows interfaces and robots running ROS. It was then incorporated into Unity once we began developing for virtual reality. ROS.NET allows for communication with ROS without an intermediary node or tunnel, messages are sent directly to and from each node to Unity. With this approach, messages are only serialized once on the Windows machine running Unity, before being recieved directly by the ROS nodes.

## 3 BENCHMARKING SETUP

While there are no comprehensive benchmarks for all of the tools available, there have been a few that we will use for our model here. Mizuchi and Inamura[24] evaluated sending messages ranging from 5k bytes to 28k bytes, with 1 to 8 clients, comparing the standard JSON approach from rosbridge to their own re-implementation. Because our primary interest is analysing applications on the VR/AR domain, we chose to scale up the data sizes to be more in line with expected robotics interfaces. We settled on 10mb, 100mb, and 1000mb sizes for our test to get a rough profile for how the amount of data affects performance. The 10mb case represents a fairly light data stream such as some robot state data. The 100mb size represents some sensor data such as a compressed image stream, a sparse pointcloud, along with some robot state data. Finally the 1000mb size is the largest data stream most networks can handle over ethernet and would represent 2-3 fairly dense point clouds, several camera streams, along with other robot information.

For the benchmarking, we used relatively high end machines running a i7 9800k Intel processor and 16GB 3300MHz RAM. In every test case, the computers were running on a gigabit network. One computer was set up to run Unity 2022.2.0b10 on Windows 10. The Unity code used in our evaluations can be found on our

Github [1]. We also had two additional computers that ran Ubuntu 18.04 with ROS Melodic. Each evaluation either used one or both of these computers, depending on the condition. The C++ code running on the ROS machines can be found on our Github [2].

## 4 EVALUATION METHODOLOGY

### 4.1 Latency: Single PC - Single PC

For the performance evaluation, we chose to perform several tests. First we look at latency, or the difference in time from when the message was sent and when the message was fully received. Each message sent has a timestamp embedded into the message, sent from Unity. Then a subscriber is setup using the roscpp C++ library, that checks the timestamp versus the current time to compute the delay. This delay is then averaged over 10 messages to get an average delay. For this test, the messages were published at 60hz. This evaluation used a single PC to single PC communication as seen in Figure 1A. and 1B.

### 4.2 Latency: Single PC - Multiple PCs

One major design trade-off of the Unity-ROS implementations is regarding the serialization of the ROS messages as described in Section 2. To analyze how latency is impacted, we repeated the latency test, however this time there was a third computer added to the setup as seen in Figure 1C. and 1D. In this single PC to multiple PCs setup, Unity first publishes the message as it did in the first evaluation. Then, for ROS-TCP-Connector and ROS#, the packets travel to Computer B which ran the roscore as well as the intermediary nodes for ROS-TCP-Endpoint and rosbridge respectively. Finally, the message is sent to a subscriber that is running on Computer C. The approach is different for ROS.NET where the message is directly sent from the Computer A to Computer C. This test was specifically looking to see if the additional network hop required by ROS# through rosbridge and ROS-TCP-Connector through ROS-TCP-Endpoint would have a significant impact on performance, or if the effect was small enough to be negligible.

### 4.3 Publishing Rate: Single PC - Single PC

We also chose to examine the publishing rate, or the maximum rate that messages could be published under each condition. This is important for systems that may be sending lots of messages, such as transforms or sensor data. An example application for this would be simulating a robot within Unity. For this test, messages were published at the fastest rate possible on each system for each message size. This evaluation used a single PC to single PC setup, where communication was between a node on Unity, and between a node running using the roscpp C++ library. The C++ node only recorded the rate messages were received over a 10 second window.

### 4.4 CPU Utilization: Single PC - Single PC

Finally we recorded the CPU utilization while running the publishing rate test. This was to determine if there was any significant difference in CPU usage between the three approaches. One potential source for differences in CPU utilization is due to the

|  | ROS-TCP-Connector | ROS# | ROS.NET |
|---|---|---|---|
| Publishing 10mb | **258.1ms** | 261.2ms | 271.5ms |
| Subscribing 10mb | **259.7ms** | 260.5ms | 272.4ms |
| Publishing 100mb | **257.6ms** | 307.7ms | 273.2ms |
| Subscribing 100mb | **254.2ms** | 309.2ms | 272.7ms |
| Publishing 1000mb | 302.4ms | 680.0ms | **280.1ms** |
| Subscribing 1000mb | 304.6ms | 679.7ms | **281.7ms** |

Table 2: Latency: Single PC - Single PC. The best performing Unity-ROS implementation for each condition is bold and highlighted.

serialization trade-off. ROS.NET handles all of the serialization on the computer running the Unity application whereas for ROS# and ROS-TCP-Connector serialization is offloaded to another computer and is handled by the intermediary nodes, rosbridge and ROS-TCP-Endpoint respectively. As the publishing rate is sending the maximum amount of messages possible, any differences in CPU usage should be most noticeable there. We chose to measure both a Unity publisher, and a Unity subscriber to see if there was a difference. We repeated each benchmark test five times and averaged the results, to smooth out any spontaneous inconsistencies.

## 5 ANALYSIS

### 5.1 Evaluation 1: Latency: Single PC - Single PC

Table 2 shows the latency of the different Unity-ROS implementations in the single PC to single PC scenario. As seen in the table, all three approaches have similar latency when the message size is small, with ROS-TCP-Connector having the best performance. When the message size becomes larger however, both ROS-TCP-Connector and ROS# gain latency, causing ROS.NET to perform comparatively better. This shows that the additional translation step in ROS-TCP-Connector did not seem to have a significant effect on performance when the message size is smaller, but may affect it for very large messages.

### 5.2 Evaluation 2: Latency: Single PC - Multiple PCs

Table 3 shows the repeated latency test with the ROS node running on a second computer. From these results we do not see a significant increase in latency, except for ROS-TCP-Connector on the large packet size test case, where we do see a significant increase. We believe this difference is explained by ROS-TCP-Connector and ROS# having an additional network hop due to ROS-TCP-Endpoint and rosbridge respectively. Since ROS.NET works without an intermediary node, and instead connects directly with each ROS node directly, and was not as heavily affected. This reinforces the previous test, that for most data sizes ROS-TCP-Connector has the superior performance, but is more heavily impacted by extremely large message sizes. Future work should investigate running this evaluation with more than one subscriber machine, as would be found in multi-robot and swarm scenarios. It would additionally be interesting to further investigate performance cross-network, this would have applications to remote teleoperation. We hypothesize

---

[1]https://github.com/uml-robotics/vamhri2023-benchmarking-unity
[2]https://github.com/uml-robotics/vamhri2023-benchmarking

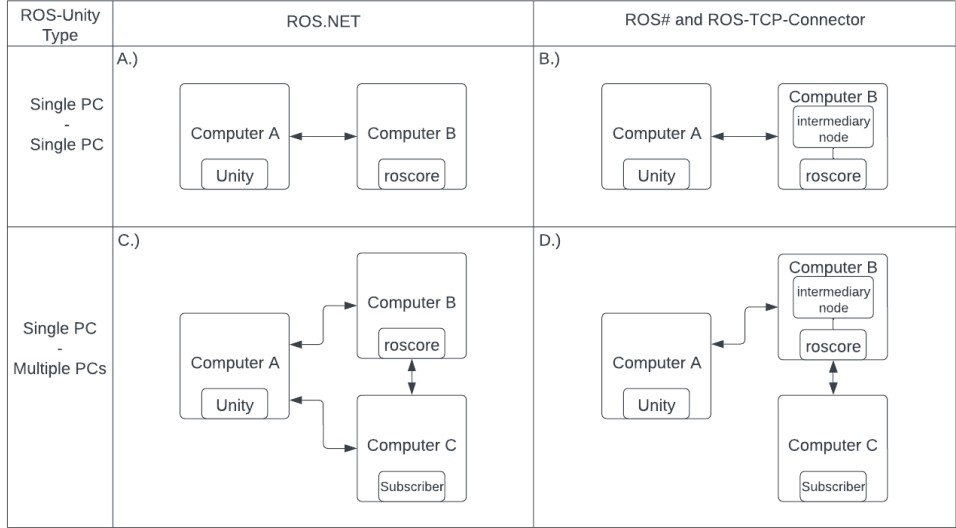

**Figure 1: Communication setups for Single PC - Single PC and Single PC - Multiple PCs across the three different Unity-ROS implementations.**

| | ROS-TCP-Connector | ROS# | ROS.NET |
|---|---|---|---|
| Publishing 10mb | **258.5ms** | 259.2ms | 273.4ms |
| Subscribing 10mb | **256.1ms** | 260.5ms | 273.6ms |
| Publishing 100mb | **256.4ms** | 314.2ms | 274.1ms |
| Subscribing 100mb | **256.8ms** | 318.2ms | 273.7ms |
| Publishing 1000mb | 411.6ms | 682.3ms | **284.7ms** |
| Subscribing 1000mb | 408.1ms | 694.3ms | **282.2ms** |

**Table 3: Latency: Single PC - Multiple PCs. The best performing Unity-ROS implementation for each condition is bold and highlighted.**

| | ROS-TCP-Connector | ROS# | ROS.NET |
|---|---|---|---|
| Publishing 10mb | **22.4%** | 22.6% | 23.1% |
| Subscribing 10mb | 22.3% | **22.1%** | 22.3% |
| Publishing 100mb | 21.8% | **21.7%** | 22.7% |
| Subscribing 100mb | 22.4% | 23.1% | **22.4%** |
| Publishing 1000mb | 23.4% | **22.3%** | 24.8% |
| Subscribing 1000mb | 22.9% | 22.4% | **22.2%** |

**Table 5: CPU Utilization During Uncapped Publishing: Single PC - Single PC. The best performing Unity-ROS implementation for each condition is bold and highlighted.**

| | ROS-TCP-Connector | ROS# | ROS.NET |
|---|---|---|---|
| Publishing 10mb | 480.2hz | 50.0hz | **531.7hz** |
| Subscribing 10mb | 482.7hz | 50.0hz | **531.2hz** |
| Publishing 100mb | **460.6hz** | 27.7hz | 416.8hz |
| Subscribing 100mb | **461.9hz** | 28.1hz | 417.6hz |
| Publishing 1000mb | 118.6hz | 2.4hz | **161.7hz** |
| Subscribing 1000mb | 116.3hz | 2.2hz | **161.2hz** |

**Table 4: Uncapped Publishing Rate: Single PC - Single PC. The best performing Unity-ROS implementation for each condition is bold and highlighted.**

### 5.3 Publishing Rate: Single PC - Single PC

The uncapped publishing rate in Table 4 shows each Unity-ROS implementation publishing as many messages as possible at various message sizes. As seen in this table the ROS# with rosbridge implementation was significantly slower at all payload sizes, and especially so when handling larger messages as we had expected. ROS-TCP-Connector and ROS.NET were similar in performance, with ROS-TCP-Connector slightly outperforming ROS.NET in the 100mb condition. ROS.NET slightly outperformed ROS-TCP-Connector in the 10mb and 1000mb condition.

### 5.4 CPU Utilization: Single PC - Single PC

Figure 5 shows the CPU overhead cost associated with each Unity-ROS approach, determined on the machine running Unity. We thought there could be a difference between CPU usage between the approaches as described in Section 4.4, however none of the approaches were CPU bottlenecked. On these relatively high end machines we were unable to determine any significant difference.

that the performance difference between ROS.NET and the other implementations will be more exaggerated in these cases.

# 6 CONCLUSION

Our goal was to determine which of the widely used Unity-ROS approaches were most prepared to handle different sized data streams. We wanted to provide that information to others preparing their own VR/AR projects so they may make an informed choice.

Overall, ROS-TCP-Connector proved to have the lowest latency in single PC to single PC and single PC to two PC for all but the largest data test, where ROS.NET has an advantage. This suggests that the serialization process for ROS-TCP-Connector is more efficient, however republishing large messages adds a significant amount of time to the latency, likely due to the intermediary node. In our single PC to two PC setup, ROS-TCP-Connector had noticeably worse results in the 1000mb condition compared to ROS.NET, which had the lowest latency in this evaluation. This suggests that projects such as multi-robot control may consider using ROS.NET.

The publishing rates between ROS-TCP-Connector and ROS.NET are similar for all message sizes, with both being affected by message size. ROS# with rosbridge performs significantly worse than ROS-TCP-Connector and ROS.NET in the uncapped publish rate scenario. Thus, any application that requires continuous publishing of lots of data is not suited for this implementation. ROS# with rosbridge generally did not perform as well as ROS-TCP-Connector and ROS.NET, however, it is still usable for most projects. The ROS# community and the popularity of this approach might also make this implementation appealing to some developers.

We were unable to determine any significant effect on CPU performance that would influence the choice. We expected performance to be impacted based on the message size, however, this was only significantly apparent in the ROS# with rosbridge publishing rate test, in all other conditions there was little difference between the 10mb and 1000mb sizes. We also found that there was not a noticeable difference between Unity providing the publisher versus the subscriber.

To summarize, we would recommend that new projects should use ROS-TCP-Connector for most applications. In multi-robot applications, or applications where you will be transmitting very large data streams to several different computers, ROS.NET would be our recommended implementation. We hope these results will prove useful to groups within the VR/AR community.

# 7 FUTURE WORK

We benchmarked performance metrics for a few situations that we feel are common in the VR/AR robotics space, specifically looking at sending a large number of messages, or sending large amounts of data at a consistent rate. However there are a number of things we didn't test for, such as message reliability, sending messages over poor network conditions such as inconsistent WiFi, or looking at swarm robot situations where there may be many different publisher/subscribers machines.

We benchmarked on relatively high end machines, as most VR/AR applications tend to require, however it's possible that very low powered machines, such as microcontrollers or a tablet, could experience very different results. Future work can further investigate the performance of Unity-ROS implementations in these niche scenarios.

# ACKNOWLEDGMENTS

This work was supported in part by the National Science Foundation (IIS-1944584) and the Office of Naval Research (N00014-21-1-2582 and N00014-18-1-2503). The views, opinions, and/or findings expressed are those of the authors and should not be interpreted as representing the official views or policies of the Department of Defense or the U.S. Government.

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
