# OpenReview forum: "Comparing Performance between different implementations of ROS for Unity"
_humanrobotinteraction.org/HRI/2023/Workshop/VAM-HRI — VAM-HRI 2023 Oral_

### Official Review · Program_Chairs · 2023-02-25
**Accept**

**Rating:** 7
**Confidence:** 5

**Review:**

Review 1:

This work investigates various performance metrics that affect Unity to ROS communication. The authors compare these metrics on three of the most commonly used communication implementations ROS-TCP-Connector, ROS#, and ROS.Net. The authors present their findings on latency, publishing rate, and CPU Utilization and the effects on various computer network setups.

Strengths:

Survey: The authors approach for surveying the number of papers that used various communication protocols seemed sound. It was interesting to see the abundance of ROS# users since the release of ROS-TCP-Connector.

Questions for the Authors:
- Why weren't any of the benchmarks run using connections running natively on a HoloLens of Magic Leap. While high end computers can handle varying loads, what about the VAM devices?

- The authors mention message reliability was not measured. Looking at the github repo for ROS.NET, the README says it is not stable. So does better latency for larger messages in ROS Net mean lower message reliability?

- An analysis on why ROS# is used 15 times to ROS-TCP-Connector 5 times since 2020 would be interesting. ROS-TCP-Connector is better but why aren't people using it? Are there applications that need ROS#? Although, this is probably a question they can't answer.

- From my understanding, only one publisher and subscriber were used at a time. However, applications typically have anywhere from 10 to 50 or more publishers. Do these benchmarks still hold in those cases?

The authors will benefit from talking about their current and future work at VAM-HRI, therefore I would argue for it to be accepted.

--------

Review 2:
 In this paper, the authors compare three different Unity-ROS implementations based on performance. I think this is an interesting foundational paper that could be of use to many researchers and would be a good fit for the VAM-HRI workshop. The paper was well written and the public code is appreciated!

I agree with Review 1 with the questions to the authors and suggestions to improve the paper. Additionally, I have a few small notes:
- on page 2, "we chose to scale up the data sizes to be more in line [with] expected robotics interfaces" seems to be missing a word.
- for new researchers, it would be helpful to describe in what scenarios one might encounter a 100mb or 1000mb message

---

### Decision · Program_Chairs · 2023-03-02

Accept (Oral)